# Changes in the Ocular Parameters of Patients with Graves’ Disease after Antithyroid Drug Treatment

**DOI:** 10.3390/medicina57050414

**Published:** 2021-04-25

**Authors:** Je-Sang Lee, Dong-Ju Yeom, Seung-Kwan Nah, Bo-Yeon Kim, Sun-Young Jang

**Affiliations:** 1Department of Ophthalmology, Soonchunhyang University Bucheon Hospital, Soonchunhyang University College of Medicine, 170, Jomaru-ro, Bucheon 14584, Korea; jesangco@hanmail.net; 2Gangnam Smile Eye Clinic, 6F, Tongyeong Building, 405, Gangnam-daero, Seocho-gu, Seoul 06615, Korea; dong_ju2000@hanmail.net; 3Department of Ophthalmology, Kim’s Eye Hospital, Myung-Gok Eye Research Institute, Konyang University College of Medicine, Seoul 07301, Korea; napole89@naver.com; 4Department of Internal Medicine, Division of Endocrinology and Metabolism, Soonchunhyang University Bucheon Hospital, Soonchunhyang University College of Medicine, 170, Jomaru-ro, Bucheon 14584, Korea; byby815@schmc.ac.kr

**Keywords:** ocular parameters, change, antithyroid drug

## Abstract

*Background and Objectives:* To find the differences in ocular axial length, keratometric measurements, and intraocular lens (IOL) power in patients with Graves’ disease (GD) after treatment with a thionamide antithyroid drug (ATD), methimazole. *Materials and Methods:* The medical charts of 28 patients (4 males and 24 females; mean age: 47.2 ± 21.2 years) were studied. Each patient was examined twice using an IOL Master Device and keratometry at the first visit (before ATD treatment) and after 1 month of ATD treatment. The IOL power was calculated for each patient using the Hoffer Q, SRK-2, and SRK/T formulas according to axial length. *Results:* After 1 month, the axial length increased (right and left eyes: *p* < 0.001 and *p* = 0.05, respectively). Based on keratometry, changes in the horizontal and vertical optical power [in diopters (D)] were not statistically significant. However, the IOL power changed after 1 month of ATD treatment in 64.3% of the patients. In 14 patients (50%), there was a 0.5–1.0 D IOL power decrease in single eyes; in two patients (7.1%), an IOL power decrease of 0.5–1.0 D in both eyes; and in two patients (7.1%), a 0.5 D IOL power increase in single eyes. The calculated IOL power values were lower after ATD treatment (right and left eyes, *p* = 0.010 and *p* = 0.018, respectively). *Conclusions:* The IOL power changed in 64.3% of GD patients after ATD treatment. Therefore, avoiding cataract surgery at the early stage of ATD treatment would be appropriate for selecting a more accurate IOL power.

## 1. Introduction

Hyperthyroidism is a pathological condition involving excess thyroid hormone that is synthesized and secreted by the thyroid gland. It is characterized by a normal or high radioactive iodine uptake by the thyroid [1,2]. The most common cause of hyperthyroidism is Graves’ disease (GD) [1], which affects 1% to 2% of the adult population [3]. The patient’s quality of life decreases because of the adverse metabolic effects of elevated thyroid hormone and thyrotropin receptor antibodies [4], which can cause emotional and sleep disturbances, and result in cosmetic effects such as a goiter and thyroid eye disease (TED) [5,6,7].

Because it has been established that the receptors of thyroid hormones are present in all tissues and organs of the body except the brain, spleen, and testicles, it is expectable that ocular biometry which is determined by the tonicity of ciliary muscles also depends on the activity of thyroid function [8]. Inspecting the previous reports, hyperthyroidism can cause refractive changes, mostly those of myopia [9,10]. However, only a few studies have reported refractive changes in TED and/or GD. There is only one report describing the relationship between TED and refractive changes. Chandrasekaran et al. [11] reported that a hypermetropic shift occurred in progressive TED. After orbital decompression, a myopic shifting occurred [11]. In their study, magnetic resonance imaging was used to assess flattening of the posterior pole as a cause of the acquired hypermetropic shift during progressive TED [11].

Treatment options for GD include antithyroid drugs (ATDs), radioactive iodine therapy, and surgery [1]. Thionamide ATDs are actively transported into the thyroid, where they inhibit iodide oxidation and organification by inhibiting thyroid peroxidase and the coupling of iodotyrosines, thereby leading to the synthesis of thyroxine (T4) and triiodothyronine (T3) [12]. Many previous studies have suggested that ATDs have immunosuppressive effects, but whether these effects involve a direct or indirect mechanism is controversial [12,13]. Immunosuppression can affect the ocular surface condition, for example by modifying the ocular surface microbiome [14]. We hypothesize that ATD treatment of GD might have an impact in ocular parameters.

In the present study, we aimed to analyze the influence of ATD of GD on ocular parameters, which has not been investigated so far. We measured the axial length, keratometric measurements, and intraocular lens (IOLs) power in GD patients after 1 month of ATD (methimazole) treatment.

## 2. Materials and Methods

This was a retrospective, observational, intra-individual, comparative study.

The medical charts of patients who were visited between September 2015 and June 2016 at a single institution were reviewed. Patients who were first diagnosed with GD at our endocrinology department were included in the study; those who had received previous ATD were excluded from the review.

All patients were evaluated by one oculoplasty specialist (S.Y.J.) to determine the degree of ocular involvement. The presence of proptosis, lid retraction, diplopia and lid swelling, clinical activity score (CAS), and conjunctival chemosis were investigated. The results of a thyroid function test were also reviewed in all subjects.

An IOL master (Carl Zeiss Meditec, Jena, Germany) was used to measure the axial length and keratometry was used to calculate the IOL power in all patients. All measurements were taken by the same technician who was blinded to the study design. The patient visits involved a pretreatment visit and a visit 1 month after ATD treatment. IOL power was decided by one surgeon (SYJ) under the assumption that cataract surgery was performed [15]. The IOL power was calculated with an A-constant of 118.7 mm (Acrysof IQ, Alcon Laboratories, Inc., Fort Worth, TX, USA) using the Hoffer Q, Sanders-Retziaff-Kraff (SRK)-2, and SRK/T formulas, according the the axial length [15,16]. For eyes with an axial length of less than 22.0 mm, the Hoffer Q formula was used [17]. The SRK-2 formula was applied to eyes with an axial length of 22.0–24.5 mm [16] and the SRK/T formula was used for eyes with an axial length exceeding 24.5 mm [18,19].

Data were available from all 28 patients in the study. The *p*-value was calculated using the Wilcoxon signed rank test using using statistical software IBM SPSS version 26.0 (IBM Corp., Armonk, NY, USA). A *p*-value of less than 0.05 was considered to indicate statistical significance.

This study conformed to the ethical guidelines of the Declaration of Helsinki and was approved by the Institutional Review Board (IRB) of the Soonchunhyang University Bucheon Hospital (IRB No. 2019-11-013, 12 February 2020).

## 3. Results

This study involved 28 patients (4 males and 24 females). The patients’ demographic characteristics are summarized in Table 1. The mean patient age was 47.2 ± 21.2 years (range: 19–77 years). There were 18 patients (64.3%) who already had TED at the first visit. The most common symptom of TED was proptosis (12 patients); other symptoms included lid retraction (8 patients), lid swelling (6 patients), and chemosis (2 patients).

The results of thyroid function tests for all 28 patients are as follows. The pretreatment averages of serum T3 and serum free T4 increased, while serum thyroid-stimulating hormone (TSH) decreased. The posttreatment averages of serum T3 and serum free T4 were in the normal range, but the posttreatment average for serum TSH decreased. Before ATD treatment, the average serum T3 was 271.73 (60–190), the average serum free T4 was 3.71 (0.89–1.78), the average serum TSH was 0.01 (0.25–4.0), and the average serum TSH-binding inhibitor immunoglobulin (TBII; thyroid-stimulating immunoglobulin) level was 15.87 (0–1.0). After 1 month of ATD treatment, the average serum T3 level was 137.85 (60–190), the average serum free T4 level was 1.67 (0.89–1.78), the average serum TSH level was 0.15 (0.25–4.0), and the average serum TBII level was 12.08 (0–1.0). CAS was not significantly different between pre and post ATD treatment (Table 2). In the present study, systemic steroid treatments were applied only in two patients who exhibited active TED at their first visit to the ophthalmology department. Several studies have attempted to evaluate the effects of additional steroid use on ATD treatment in GD patients, due to the autoimmune nature of GD [20]. A recent meta-analysis demonstrated a strong reduction in the recurrence risk of GD when immunosuppressive drugs were added to ATD treatment in GD patients [21].

After 1 month of ATD treatment, the axial length increased (Table 3). At the first visit, the mean axial length was 24.38 ± 1.54 mm in the right eye and 24.43 ± 1.48 mm in the left eye. After 1 month of treatment, the mean axial length was 24.40 ± 1.54 mm (*p* < 0.001) in the right eye and 24.46 ± 1.48 mm (*p* = 0.005) in the left eye.

Keratometry of the right eye showed that the horizontal power (D) was 42.99 ± 1.13 D at the first visit and 43.01 ± 1.09 D 1 month later (*p* = 0.084); the vertical power was 43.54 ± 1.51 D at the first visit and 43.61 ± 1.48 D 1 month later (*p* = 0.616). In the left eye, the horizontal power was 42.97 ± 1.06 D at the first visit and 42.98 ± 1.04 D 1 month later (*p* = 0.612); the vertical power was 43.48 ± 1.63 D at the first visit and 43.51 ± 1.44 D 1 month later (*p* = 0.891) (Table 3).

The calculated IOL power values were lower after ATD treatment (right and left eyes, *p* = 0.010 and *p* = 0.018, respectively; Table 3).

The IOL power, as calculated using the Hoffer Q, SRK-2, and SRK/T formulas, also changed after 1 month of ATD treatment (Table 4), although no changes were observed in 10 patients (35.7%). However, in 14 patients (50%), there was a 0.5–1.0 D decrease in single eyes; in two patients (7.1%), there was an IOL power decrease of 0.5–1.0 D in both eyes; and in another two patients (7.1%), there was a 0.5 D increase in single eyes. Overall, 18 patients (64.3%) experienced a change in IOL power after ATD treatment.

## 4. Discussion

In the present study, we compared the axial length, keratometric measurements, and IOL power in GD patients after ATD treatment. The results show that the axial length increased and the IOL power changed in 64.3% of acute-phase GD patients. To our knowledge, this is the first report to investigate the influence of methimazole of GD on ocular parameters. However, inspecting our results, the change of axial length after ATD treatment was subtle and the changes in IOL power was 0.2 D for the right eyes and 0.18 D for left eyes. When presuming the IOL is only available in 0.5 D increments, the change in 0.2 D might not be relevant to use of the IOL. Thus we further analyzed the number of patients who experienced the IOL changes after ATD treatment. Eighteen patients (64.3%) experienced a change in IOL power after ATD treatment. These results may have clinical significance for cataract surgeons. Avoiding cataract surgery at the early stage of ATD treatment would be appropriate for more accurate IOL power selection.

Previous studies have reported that some systemic diseases and treatment with certain medications, such as topiramate and acetazolimide, result in myopic changes [22,23,24,25]. Although the exact mechanism of myopic change is unknown, the anatomical changes involve the release of posterior pole flattening, forward movement of the lens-iris diaphragm, and thickening of the lens, resulting in a change in refractive power, ciliary muscle edema, and twitching of the choroidal exudate, resulting in myopic changes [11,25,26,27].

Methimazole is an ATD that inhibits the enzyme thyroperoxidase. Thyroperoxidase participates in thyroid hormone synthesis by oxidizing the anion iodide to molecular iodine, hypoiodous acid, and enzyme-linked hypoiodate, which facilitate the addition of iodine to tyrosine residues in the hormone precursor thyroglobulin. This is a necessary step in the synthesis of T3 and T4 [12]. In addition, Cooper [12] and Antonelli et al. [13] reported that ATD may also have anti-inflammatory and immunosuppressive effects. In the present study, we hypothesize that ATD treatment can induce changes in ocular parameters in patients with GD. As a result, we found that the axial length increased and the IOL power changed in 64.3% of acute-phase GD patients. In total, 50% of patients exhibited TED during their first visit. However, no changes in CAS were observed after ATD treatment. Thus, the possibility that previous treatment for TED and/or changes in the clinical activity of TED could affect ocular parameters was ruled out.

Chandrasekaran et al. [11] reported the presence of acquired hypermetropia during progressive GD. They hypothesized that a combination of enlarged extraocular muscles, anterior displacement of the globe, and orbital hypertension were related to the elevated muscle and fat volumes in flattening of the posterior pole during GD [11]. After orbital decompression, a significant myopic shift was observed [11].

The exact mechanism of the increased axial length and changes in the intraocular D at 1 month after ATD treatment could not be determined from the present study, but we assumed that these changes resulted from the anti-inflammatory and immunosuppressive effects of the ATD, resulting in enlarged extraocular muscles and reduced retrobulbar fat volumes. The eyeball then undergoes a decrease in posterior pole flattening, followed by elongation of the axial length.

The present study has a limitation that should be addressed. Notably, the axial length, keratometry, and IOL power calculation were established as the outcome measures. IOL power was decided by one surgeon under the assumption that cataract surgery was performed. This is because we wanted to intuitively show the clinical importance of refractive error changes in GD patients after ATD treatment in order to facilitate reader comprehension. However, IOL power selection is more complicated in clinical practice [28,29,30,31,32]. To ensure accurate prediction of IOL power, vergence lens formulas have incorporated additional biometric variables, including anterior chamber depth, lens thickness, white to white measurement, and age [33]. Various formulas, such as Haigis, SRK/T, Holladay 2, or Hoffer Q can be used depending on the subjects’ characteristics.

## 5. Conclusions

In summary, in patients with acute-phase GD, the axial length increased after 1 month of ATD treatment. In addition, the IOL power, calculated using the Hoffer Q, SRK-2 and SRK/T formulas, was changed in 64.3% of patients. This study had some limitations in that it was not a case-controlled study, the sex ratio of the subjects was not balanced, and the sample size was small. Nonetheless, this study characterized a novel drug interaction of an ATD in the eye for the first time. Additional case-control studies could provide insight into the mechanism of action of ATD. Furthermore, avoiding cataract surgery in the early stage of ATD treatment would be appropriate for selecting a more accurate IOL power.

## Figures and Tables

**Table 1 medicina-57-00414-t001:** Patient demographics.

Demographic	Score
Total	28 (100.0)
Male	4 (14.3)
Female	24 (85.7)
Thyroid eye disease at first visit	
Positive	18 (64.3)
Negative	10 (35.7)
Thyroid eye disease symptom	
Proptosis	12
Lid retraction	8
Lid swelling	6
Chemosis	2
Age (years)	47.2 ± 21.2 (19–77)
Smokers	5 (17.9)
Thyroid eye disease treatment	
Steroid (systemic)	2
Transconjunctival triamcinolone injection for upper eyelid retraction	2
Radiation	0
Surgery	0

The data is presented as *n* (%) or mean ± standard deviation (range).

**Table 2 medicina-57-00414-t002:** Thyroid function test and clinical activity score between pre- and post-antithyroid drug treatment of the subjects.

Variable	Pre-Antithyroid Drug	Post-Antithyroid Drug	*p* Value
Serum triiodothyronine (T3) (60–190)	271.73	137.85	<0.01
Serum free thyroxine (T4) (0.89–1.78)	3.71	1.67	<0.01
Serum thyroid stimulating hormone (TSH) (0.25–4.0)	0.01	0.15	<0.01
Clinical activity score (CAS)	1.9 ± 0.9	1.6 ± 0.6	0.09

**Table 3 medicina-57-00414-t003:** Comparison of axial length and keratometry measurements between pre- and post-methimazole treatment of Graves’ disease patients.

Variable		Measurements		
		Pre	Post	*p* Value *
In the right eyes	Axial length (mm)	24.38 ± 1.54	24.40 ± 1.54	<0.001
	Keratometry measurements (D)			
	Horizontal	42.99 ± 1.13	43.01 ± 1.09	0.804
	Vertical	43.54 ± 1.51	43.61 ± 1.48	0.616
	Degree (0–360°)			
	Horizontal-axis	76.57 ± 67.97	83.96 ± 66.79	1
	Vertical-axis	94.07 ± 42.21	88.61 ± 42.13	0.193
	Intraocular lens (IOL) power (D)	19.50 ± 4.80	19.30 ± 4.86	0.010
In the left eyes	Axial length (mm)	24.43 ± 1.48	24.46 ± 1.48	0.005
	Keratometry measurements (D)			
	Horizontal	42.97 ± 1.06	42.98 ± 1.04	0.612
	Vertical	43.48 ± 1.63	43.51 ± 1.44	0.891
	Degree (0–360°)			
	Horizontal-axis	71.64 ± 65.82	78.79 ± 62.76	0.819
	Vertical-axis	110.21 ± 33.74	104.5 ± 40.14	0.14
	IOL power (D)	19.38 ± 4.68	19.20 ± 4.59	0.018

The data is presented as mean ± standard deviation. D: diopter. * *p*-value were calculated by Wilcoxon signed rank test.

**Table 4 medicina-57-00414-t004:** Intraocular lens power difference (using the Hoffer Q, SRK-2, and SRK/T formulas) between pre-and post-methimazole treatment of Graves’ disease patients.

Variable	Patients No (%)
Total	28 (100.0)
No difference on both eye	10 (35.7)
Difference in single eye (+0.5D)	2 (7.1)
Difference in both eye (−0.5~−1.0D)	2 (7.1)
Difference in single eye (−0.5~−1.5 D)	14 (50)
Difference in right eye (−0.5~−1.0 D)	6 (21.4)
Difference in left eye (−0.5~−1.5 D)	8 (28.6)

The data is presented as *n* (%). D: diopter.

## Data Availability

The data collected and analysed in the study will be available from the corresponding author on reasonable request.

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
