# Peer review of "Changes in the Ocular Parameters of Patients with Graves’ Disease after Antithyroid Drug Treatment"

_medicina, 2021, doi:10.3390/medicina57050414_

Round 1

Reviewer 1 Report

Reference 14, Petrillo et al., was deleted. It should be included because ocular surface microbiota description is missing in the manuscript. 

Author Response

Reviewer Comments:

Reviewer #1:

Reference 14, Petrillo et al., was deleted. It should be included because ocular surface microbiota description is missing in the manuscript.

Response: Reference 14 (Petrillo et al.) was deleted during the revision process. As the discussion of the microbiota contents deviates from the main theme of the paper, the contents have been removed in the revised manuscript. However, if the reviewer feels that the Petrillo et al. reference is required, we would be happy to include it in the revised manuscript.

We are grateful to the reviewer for reviewing our article. We hope that the revisions are satisfactory and that the manuscript is now suitable for publication in Medicina Journal.

Yours sincerely,

Sun Young Jang, MD.

Reviewer 2 Report

Executive Summary

The manuscript titled “Changes in the Ocular Parameters of Patients with Graves’ Disease after Antithyroid Drug Treatment” describes a pilot study using Methimazole as an intervention to treat thyroid eye disease (TED). Overall, as a pilot study, the authors invested sufficient resources to start the project. The manuscript is scientifically written with strong logic. The authors may perform minor revisions to provide more background information and further discussions to promote future research in this direction.

Major Comments

  • Since the intervention is Methimazole only and the target is thyroid eye disease, the authors may change the title to better reflect the goal of this research.

Minor Comments

  • Line# 156-160: Based on the data, there are significant changes for the axial length and the IOL power. However, the authors may need to discuss further to tell readers what is the meaning of these changes. Are they a sign of improvement? Please be advised that readers of Medicina may not be ocular experts. They may be general audiences or patients with eye diseases.
  • Line# 162: “certain medications result in myopic changes”. In the background section, the authors did not provide patients' current medication. Some patients with previous conditions, may or may not be taking similar drugs or drugs that have drug-drug-interactions with Methimazole. It would be helpful to include that information for better judgment.
  • Line# 190: there are more limitations to discuss. First of all, gender is unbalance. Further, the sample size is very limited. There are other aspects if authors wish to discuss, for example, smoking conditions, alcohol taking, illicit drug usage. However, they are recommended but not required.

Author Response

Reviewer #2:

Executive Summary

The manuscript titled “Changes in the Ocular Parameters of Patients with Graves’ Disease after Antithyroid Drug Treatment” describes a pilot study using Methimazole as an intervention to treat thyroid eye disease (TED). Overall, as a pilot study, the authors invested sufficient resources to start the project. The manuscript is scientifically written with strong logic. The authors may perform minor revisions to provide more background information and further discussions to promote future research in this direction.

Major Comments

Since the intervention is Methimazole only and the target is thyroid eye disease, the authors may change the title to better reflect the goal of this research.

Response: We have revised the title in accordance with the reviewer’s suggestions. However, the subjects in the present study were not only patients with thyroid eye disease but also patients with Graves’ disease and without thyroid eye disease. Therefore, we have changed “antithyroid treatment” to “Methimazole” in the revised title.

Title: Changes in the Ocular Parameters of Patients with Graves’ Disease after Methimazole Treatment

Minor Comments

Line# 156-160: Based on the data, there are significant changes for the axial length and the IOL power. However, the authors may need to discuss further to tell readers what is the meaning of these changes. Are they a sign of improvement? Please be advised that readers of Medicina may not be ocular experts. They may be general audiences or patients with eye diseases.

Response: There are significant changes in the axial length and the IOL power. These findings are clinically significant for cataract surgeons when selecting the accurate IOL power during cataract surgery. These findings are not a sign of improvement or deterioration. We emphasized the clinical significance of these findings as follows:

In the present study, we compared the axial length, keratometric measurements, and IOL power in GD patients before and after ATD treatment. The results showed that the axial length increased and the IOL power changed in 64.3% of acute-phase GD patients. To our knowledge, this is the first study to investigate the influence of ATD of GD on ocular parameters. These results may have clinical significance for cataract surgeons. Avoiding cataract surgery at the early stage of ATD treatment would be appropriate for more accurate IOL power selection.

Line# 162: “certain medications result in myopic changes”. In the background section, the authors did not provide patients' current medication. Some patients with previous conditions, may or may not be taking similar drugs or drugs that have drug-drug-interactions with Methimazole. It would be helpful to include that information for better judgment.

Response: We have revised the text as follows:

Previous studies have reported that some systemic diseases and treatment with certain medications, such as topiramate and acetazolamide, result in myopic changes

Line# 190: there are more limitations to discuss. First of all, gender is unbalance. Further, the sample size is very limited. There are other aspects if authors wish to discuss, for example, smoking conditions, alcohol taking, and illicit drug usage. However, they are recommended but not required.

Response: We have added limitations in accordance with the reviewer’s suggestions.

In summary, in patients with acute-phase GD, the axial length increased after 1 month of ATD treatment. In addition, the IOL power, calculated using the Hoffer Q, SRK-2, and SRK/T formulas, was changed in 64.3% of patients. This study had some limitations in that it was not a case-controlled study, the sex ratio was not balanced between groups, and the sample size was small. Nonetheless, this study characterized a novel drug interaction of an ATD in the eye for the first time. Additional case-control studies could provide insight into the mechanism of action of ATD. Furthermore, avoiding cataract surgery in the early stage of ATD treatment would be appropriate for more accurate selection of IOL power.

We are grateful to the reviewer for reviewing our article. We hope that the revisions are satisfactory and that the manuscript is now suitable for publication in Medicina Journal.

Yours sincerely,

Sun Young Jang, MD.

Round 2

Reviewer 1 Report

In my opinion, microbiota surface discussion has to be included in the introduction. 

Author Response

Reviewer #1:
In my opinion, microbiota surface discussion has to be included in the introduction.

Response: We appreciate the positive comments by Reviewer #1, as well as this suggestion. We included the above mentioned reference in the Introduction section, and have revised the text as follows:

Page 2, Introduction: Treatment options for GD include antithyroid drugs (ATDs), radioactive iodine therapy, and surgery. Thionamide ATDs are actively transported into the thyroid, where they inhibit iodide oxidation and organification by inhibiting thyroid peroxidase and the coupling of iodotyrosines, thereby leading to the synthesis of thyroxine (T4) and triiodothyronine (T3) [12]. Many previous studies have suggested that ATDs have immunosuppressive effects, but whether these effects involve a direct or indirect mechanism is controversial [12, 13]. Immunosuppression can affect the ocular surface condition, for example by modifying the ocular surface microbiome [14]. We hypothesize that ATD treatment of GD might have an important impact on ocular parameters.

Added references

[14] Petrillo, F.; Pignataro, D.; Lavano, M.A.; Santella, B.; Folliero, V.; Zannella, C.; Astarita, C.; Gagliano, C.; Franci, G.; Avitabile, T.; et al. Current Evidence on the Ocular Surface Microbiota and Related Diseases. Microorganisms 2020, 8.

Yours faithfully,

Sun Young Jang, MD.